# A Laboratory-Scale Study of Selected Chinese Typical Flammable Wildland Timbers Ignition Formation Mechanism

Wenxu Yang [1,2], B. H. Abu Bakar [1,*], Hussin Mamat [3], Liang Gong [4] and Nursyamsi Nursyamsi [5]

1   School of Civil Engineering, Engineering Campus, Universiti Sains Malaysia, Penang 14300, Malaysia
2   Chongqing Dadukou District Department of Housing and Urban Rural Development,
    Chongqing 400084, China
3   School of Aerospace Engineering, Engineering Campus, Universiti Sains Malaysia, Penang 14300, Malaysia
4   Department of Fire protection Engineering, Southwest Jiaotong University, Chengdu 610032, China
5   Department of Civil Engineering, Faculty of Engineering, Universitas Sumatera Utara, Padang Bulan,
    Medan 20155, Indonesia
*   Correspondence: cebad@usm.my

**Abstract:** Firebrands are the primary source of ignition for large wildfires and urban wildfires (WUIs). China is a country with a high incidence of forest fires, and there are great differences in the terrain, climate, and other natural conditions in different regions; the frequency of forest fire will lead to greater regional differences. In the process of fighting forest fire, the fire commander should make an accurate analysis and judgment according to the various signs of the fire, which are the key to ensure the safety of the participants and to realize a quick decision. Existing studies of firebrands formation have been performed using limited quantities of wildland fuels with limited MC fuel levels and environmental conditions and lacking comprehensive data analysis including typical wildland timbers basic fuel, pyrolysis and flammability properties, and forest fire dynamic knowledge (including forest fire development period analysis and the harm of heat flux to the human body) to guide the firefighting strategy. To better understand the characteristics of firebrand formation in different Chinese regional places, a systematic study to quantify wildland fuels ignition formation by testing different fuels under different conditions is needed. The objective of this study was to determine the basic pyrolysis and flammability of wildland fuels with high fire intensity in typical areas of China to provide relevant property data, offering insight into how wildland fuels arrangement can determine the movement of wildfires for firefighting strategy. Thermogravimetric analysis (TGA) was used to determine the pyrolysis performance of selected wild fuels under different heating rates and different fuel MC values. The flammability of selected wildland fuels at different heat fluxes and at different moisture contents was determined using a cone calorimeter. This study measured the pyrolysis and flammability of some selected wildland fuels and found that some controlling factors (MC levels, heating conditions) influenced the outcome variables, especially the flammability of wildland timber. Fire behavior refers to the intensity at which a fire burns and how it moves. This research results point out the following: (1) Forest fire barriers or fuel breaks should be separated among *Eucalyptus robusta Smith* and *Pinus massoniana* before or in the fire due to high risk for ignition and strong flammability, and it is suggested to remove, control, and replace high-risk flammable timbers with low-risk flammable timbers as a part of long-term wildland fire management strategies. (2) Fire commanders could utilize some research to test conclusions and make an accurate analysis and judgment: The TTI time for each material indicates the ideal time for firefighters to put out fire, the peak of heat-release time indicates a fully developed fire to suggest firefighters finish work before the forest fire spirals out of control, and the flameout time represents the moment of low risk of fuel ignition, so firefighters could allow the fuel to burn out and change the extinguishing target to other regions of developing forest firebrands.

**Keywords:** pyrolysis and flammability; high ignition possibility; quantify ember production

## 1. Introduction

### 1.1. Background of Chinese Forest Fire and Firebrand Simulation

Since 18 August 2022, mountain fires have broken out in Jiangjin, Dazu, Tongliang, and Banan District districts of Chongqing City due to persistent heat and drought. A total of more than 5000 professional rescue teams from urban and district emergency bureaus, forest fire fighting groups, armed police forces, firefighting and rescue, as well as local cadres and masses were mobilized to put out the fire. The municipal Emergency Bureau Aviation Rescue Team utilized seven helicopters to carry out air relief for the fire; a total of 540 households moved, including more than 1500 people; and more than 200,000 people were directly affected by the disaster. Due to the influence of dry and flammable vegetation, heavy fuel load, and strong wind, many fire sites rekindled and spread in different degrees.

In the process of fighting forest fire, the fire commander should make an accurate analysis and judgment according to the various signs of the fire, which are the key to ensure the safety of the participants and to realize a quick decision. Therefore, it is necessary for firefighting commanders to study the following main forest fire behavior changes and practical observation techniques: first, the change of forest fire behavior, which can easily to cause casualties; second, using smoke color to judge forest fire behavior and forest fire site topography; third, observing the fire convection column to judge the wind direction, wind speed, and distance; and fourth, forest fire safety time division. Mastering the above-mentioned forest fire behaviors and the practical observation technique of a fire scene can help fire commanders grasp a fire scene change and obtain fire scene information at any time.

The problem of habitat destruction associated with fire, known as the wildland–urban interface (WUI) fire problem, is generally defined as areas where man-made structures and infrastructure combine or mix with natural vegetation types [1]. In addition to the fact that WUI is already a global problem, wildfires pose an increasing threat to people and ecosystems, with negative impacts including property and infrastructure loss, economic disruption, ecosystem degradation, and soil erosion alongside the high cost of fighting fires [2].Unburned vegetation can be roughly described as fuel firebrand. As the fire front approaches, the particles are thermally damaged. The effects of heat transfer on fire propagation are mainly radiation and convection. At the front of the fire front, the heat transfer by radiation is greater than the heat transfer by convection. Radiation sources include fire fronts and glowing vegetation. Convective heating requires airflow from the burning zone to come into contact with unburned vegetation. (Vegetation is otherwise cooled only by convection.) When plant particles receive heat from a fire source, they heat up [3–5]. When the temperature is high enough (usually around 100 °C), drying begins. Vegetation moisture, known as fuel moisture content (FMC), plays a crucial role in fire spread as a heat sink that slows or prevents fuel burn. Pyrolysis begins when the plant pellets dry out. Nearby flames ignite the burning mixture and add to the flames. First, flammability occurs in the gas phase [6,7]. Flammability occurs when the particles leave the gas and become completely carbonized. It occurs on the surface of the coke, and the particles glow, emit a large amount of radiation, and burn slowly. After the wildland fuel is completely consumed, the particles turn into ash. Fire has three elements at once: a flammable gas, oxygen (in the air), and a source of heat sufficient to ignite the flammable mixture [8]. Due to the rich oxygen content in the forest atmosphere, wildfire spread is generally not restricted. Wildland timber is the main fuel for forest fires, and its distribution has a significant impact on heat transfer.

Flame propagation and wildland ignition can be understood as three main processes or mechanisms: firebrand generation, ember transport, and fuel flammability. There are three main mechanisms for the fire propagation of the flame [9]; forest fires involve both chemical and physical processes. When fuel is burned in the wild, its stored chemical energy is converted into heat, i.e., thermal energy, through complex chemical reactions. However, for the reaction to start, heat must be transferred from the burning cinders to

the fuel, and if the fire is to continue to burn and spread, heat must be transferred to the unburned fuel.

The main components of wood are cellulose, hemicellulose, and lignin, and the additional components of wood, such as extract and ash, have a low content in wood but have a great impact on wood combustion performance. In general, the higher the extraction content of wood, the more easily it is burned, while the ash content of wood is more difficult to burn [9]. The burning of wood is essentially the burning of the flammable products produced during the thermal decomposition of wood, and the thermal decomposition reaction of wood is essentially the sum of the thermal decomposition reaction of the three main chemical components in the cell wall of wood. Hemicellulose is the most unstable, decomposing in the temperature range of 225–325 °C; cellulose decomposes in the higher temperature range of 325–375 °C and lignin in the temperature range of 250–500 °C. Hemicellulose and cellulose break down to form a large number of volatile products, while lignin mainly forms charcoal [10–12]. The start of pyrolysis and further flammability could be delayed by the presence of water in wood due to energy loss caused by water evaporation [13]. The flammability value of wildland wood or bark depends on the amount of heat energy that can be recovered, which in turn varies according to the moisture content and chemical composition of the wood or bark. Data on wood or bark fuels are derived from different sources.

### 1.2. Experiments and Literature Review

The objective of this study was to determine the basic pyrolysis and flammability of wildland species in typical forest areas of China, including the basic pyrolysis and flammability. The productivity, quality, shape, and size of the torches are determined under a range of conditions; the duration of the flammability is determined; and the impact of these characteristics on the ignition potential and spread of the fire is assessed. In the first step, the following seven forest tree species were selected: *Pinus sylvestris* var. *Mongolica*, *Eucalyptus robusta*, *Cupressus funebris Endl*, Masson pine, larch, birch, and shiromatasu. Pyrolysis and flammability tests were then carried out: the pyrolysis performance of the selected wood species was measured by using thermogravimetric analysis (TGA) technology at different heating rates and different fuel moisture contents. Existing studies of firebrands formation have been performed using limited quantities of wildland fuels and with limited MC fuel levels and environmental conditions, lacking comprehensive data analysis including typical wildland timbers basic fuel, pyrolysis and flammability properties, and forest fire dynamic knowledge to guide firefighting strategy.

IRI researchers conducted experimental and field studies on tinder and produced three main reports. Waterman performed laboratory experiments on roof flame generation. Waterman and Tanaka studied the artificial ignition of various flammable materials based on roof fire experiments [1]. Vodvarka studied the size and location of flashpoints in building fires, wrote a report on flashpoint surveys, and investigated the experience of firefighters and firefighters in the field using flashpoints [2]. ITRI's experimental studies mainly cover three ignition mechanisms: ignition generation, ignition transfer, and ignition. The ignition of acceptor fuel is considered to be the process of energy transfer from fuel ember to fuel, which is closely related to the flammability characteristics of wildland fuel embers. Clements, with the USDA Forest Service, performed drop tests on burning fires based on terminal velocity approximations developed by Tarifa. He determined their final velocities by throwing different types of combustibles from a given height, calculating their free-fall times, and incorporating the equations of motion for objects in the air. Fourteen hardwood leaves, three pine needles, six pine cones, saw palmetto leaves, reindeer moss, Spanish moss, and paperbirch bark were tested [3–5]. In a vertical wind tunnel, four types of pine cones burned and glowed in flight. The study found that igniters that burn a high proportion of the sample during flight and landing tended to have higher terminal velocities. Pine cones glow an order of magnitude longer than they burn. Clements concluded that the size of the fuel could allow a fire to cause serious hazards. Finally, experiments on dry

fuel-bed ignitions by wildland fuels were recommended. More recently, CSIRO in Australia conducted vertical wind tunnel tests [6–8]. A vertical wind tunnel designed by CSIRO was successfully used to test the ultimate speed of isolated flammability and has two unique features. The vertical working section has a diverging cone that allows the flame to find its final velocity in the velocity gradient, making the working section have a higher boundary layer velocity than the middle to avoid this. Flames have spread to the walls of the work area. The installation is the first to study the aerodynamic properties and flammability of eucalyptus bark fuel [9–11].

The National Institute of Standards and Technology (NIST) conducted ignition tests on different types of fuel beds by means of an ignition burner. They placed burning and glowing yellow pine tinder between pine needles, beds of shredded paper, and cedar wood chips as well as pine thatch and hardwood mulch beds and mowed grass beds to see if fires occurred [12–14]. Several 25 mm or 50 mm diameter tanks were placed on each fuel bed while maintaining a thickness-to-diameter ratio of 1:3, as in the Woycheese tunnel test. A unique experimental setup called a flammability generator was developed to generate luminescent combustion with a controllable and reproducible size and mass distribution. Based on previous wildland fuels-generation studies, the size and mass distribution of flames produced by generators was chosen to represent flames produced by flammability in power plants [15–17]. The susceptibility of roof coverings to flame attack depends on the flame flow generated by the equipment. Gould et al. published a paper on the CSIRO Vesta project, which included data on spark size and travel distance for regulated fires [18]. Knowledge of ignition sources and ignition has developed over 50 years of research, most of which has focused on ignition distances. Few studies describe the ignition and subsequent ignition of building materials or plant fuels. Developing science-based mitigation strategies for WUI fires requires an understanding of the ignition process [19]. In fact, the generation of the flash point is the first step to generate the flash-point phenomenon, and it is also the basis for understanding the subsequent transfer and ignition process. Understanding the ignition process can also help improve modeling of large-scale forest fire propagation (wildland, WUI, or urban fires) and develop appropriate mitigation strategies. Flammable products are affected by many factors, including fuel type, fuel condition (live or dead fuel, moisture content, etc.), fuel thermal decomposition properties, fuel flammability properties, and environmental conditions to which the fuel is exposed (wind, relative humidity, temperature, heat conditions, etc.) [20]. These control factors affect outcome variables such as probability of ignition, rate of occurrence of ignition, physical characteristics of ignition (e.g., size, mass and shape of ignition, distance traveled), and ignitability characteristics of ignition (e.g., duration of burning, time of burning, latent heat energy, latent heat flux, and temperature).

## 2. Experiment Design and Method

### 2.1. Timber Species Selection

With the development of research on flammable trees, the factors affecting the flammability of epiphytes are attributed to the calorific value, moisture content, and oil content. Epiphytes grow mostly in a dark, damp environment. Therefore, the terrain and environment also have an important impact on the flammability of epiphytes.

To select typical flammable Chinese wildland timbers as specimens for this study, the main basis for selection was that the chosen fuels should be representative of typical wood with high ignition potential and thought to be capable of generating a large amount of firebrands in the following provinces: Guangxi, Hunan, and Guangdong, which account for 70% of national forest fire incidents according to the statistics provided by the State Forest Administration of China (Figure 1) (China Forestry Information Network).

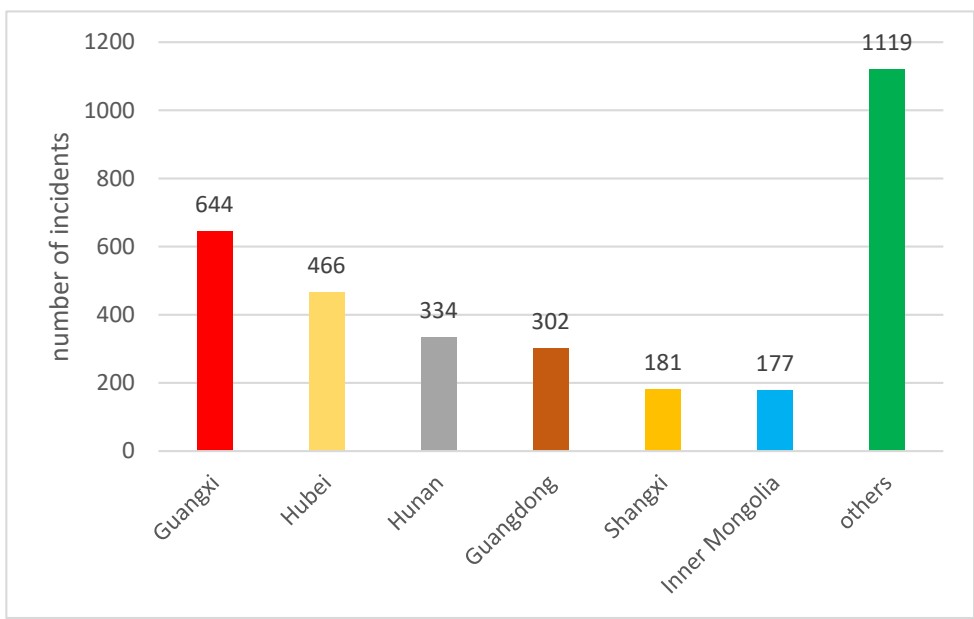

**Figure 1.** Forest Fire Incidents Statistics by the State Forest Administration of China.

In 2016, the Ministry of Forestry put forward the plan of establishing national biological fire-resistant forest belt project, which promoted the research of fire-resistant tree experiments. The list of non-flammable and flammable tree species is given in Tables 1 and 2.

**Table 1.** Non-flammable tree species.

| Southern Region | | Northern Region | |
|---|---|---|---|
| *Schima superba* | *Michelia macclurei* | *Fraxinus mandshurica* | *Ulmus propinqua* |
| *Schima argentea* | *Acaci auriculaeformis* | *Phellodendron amureense* | *Ulmus laciniata* |
| *Schima wallichiii* | *Acacia confuse* | *Acer spp.* | *Juglans mandshurica* |
| *Acaciamangium* | *Myrica rubra* | *Salix sp.* | *Larix gemelininii* |
| *Alnuscremastogyne* | *Vernicia fordii* | *Populus sp.* | *Larix spp.* |
| *Castanopsis* | *Quercus glauca* | *Malus baccata* | *Larix kaempferi* |
| *Myricarubra* | *Alnus cremastogyne* | *Tilia amurense* | |

**Table 2.** Flammable tree species.

| Species | MC Level | Wood Oil Content | Dry Weight (g) | Surface Flammable Coverage % | Hardness |
|---|---|---|---|---|---|
| Poplars | Medium | Low | 1280 | 92 | Soft |
| Birch | Medium | Low | 2320 | 91 | Hard |
| Scotch pine | Medium | High | 2670 | 95 | Soft |
| Korean pine | Medium | High | 3875 | 96 | Soft |
| Ash tree | Medium | Low | 527 | 56 | Soft |
| Walnut | High | Low | 627 | 48 | Soft |
| Phellodendron | High | Low | 725 | 58 | Hard |
| Lime tree | Medium | Low | 1050 | 72 | Soft |
| Larch | Medium | Medium | 4250 | 85 | Soft |

Apparently, pine with high wood oil content and surface flammable coverage is considered as the top flammable tree species.

Eucalyptus is extremely adaptable. They are not only cold-hardy but also heat-resistant, so eucalyptus can be grown in both tropical and boreal regions. From the 19th century, many countries began to introduce eucalyptus from Australia. By the twentieth century, more than 96 countries and regions had planted eucalyptus. Eucalyptus in my country is mainly concentrated in Guangdong and Guangxi, Yunnan, Sichuan, Fujian, and other places. Eucalyptus is the main cause for many forest fire incidents, especially in Australia. Eucalyptus contains oil, and there are aromatic oils on its leaves; eucalyptus oil can even be directly extracted, which easily catches fire. If there are some small flames around the eucalyptus, this could easily cause fires.

Based on the relevant research, seven fuels (including three pines, one eucalyptus, one Endl, one larch, and one birch) with high ignition from provinces that represent the typical forest fire incidents were used for this thesis, as shown in Table 3.

**Table 3.** Recycled flammable tree species selection.

| Designation | Description | Source Place | Hardness |
|:---:|:---:|:---:|:---:|
| A | *Pinus sylvestris var.*—hard pines | Inner Mongolia | Very hard |
| B | *Eucalyptus robusta Smith* | Guangdong | Very hard |
| C | *Cupressus funebris Endl* | Guangdong | Medium |
| D | *Betula platyphylla Suk* | Dongbei | Medium |
| E | *Larix mastersiana Rehder*—larch | Sichuan | Soft |
| F | *Pinus massoniana Lamb* | Guangxi | Soft |
| G | *Pinus bungeana* | Shuangxi | Soft |

Notes: All material will henceforth be described with its designated name; for example, material *Pinus sylvestris var.*—hard pines will be marked as material A.

### 2.2. Experiment Design

Several control factors affect the outcome variables, such as the probability, formation rate, physical properties of wood pyrolysis products (e.g., size, mass, and shape), and flammability properties of wildland timber; the objective of this work is to study the flammability of different combustible woods to quantify ignition, heat transfer, and flameout. As the temperature strongly dominates the chemical reactions, much research has investigated the effect of heat flux on the flammability of different wood species.

In order to accomplish the above objectives and explore the relationship between parameters in Table 4, the following tasks were conducted:

1. Select appropriate representative flammable wildland fuels in typical waste types in China as specimens for this study;
2. Conduct an experimental study to measure and control the moisture content of selected agricultural fuels at the project MC level;
3. Carry out TGA experiment and follow-up data analysis to obtain three preset MC levels and choose off-road fuel at the three preset levels. The main parameter of thermal decomposition rate is set below the heating rate;
4. Conduct cone calorimeter experiments and subsequent data analysis in order to obtain basic flammability properties of the selected wildland fuels at the three predetermined MC levels and three pre-determined heat-flux levels.

**Table 4.** Controlling factors and parameters.

| Properties | Controlling Factor | Relationship | Parameters |
|---|---|---|---|
| Fuel | MC level | | Thickness, area, and density |
| Pyrolysis | MC level | Increase/decrease/ non-influence | Pre-exponential factor (ln(A)), activation energy (E), and thermal conductivity |
| | Temperature | | |
| | Heating rate | | |
| Flammability | MC level | | Time to ignition (TTI), critical heat flux (CHF), peak of heat-release rate (PHRR), mass-loss rate, effective heat of flammability (EHC), and time to flameout |

### 2.3. Experiment Method

The MC levels of most wildland fuels vary in the 5% to 15% range. Nominal MC levels of 5%, 10%, and 15% were obtained using dry desorption and adsorption processes in a laboratory oven at ambient conditions, and the effect of MC levels on thermal degradation and flammability was tested. The MC of a wood or wood-based composite can be calculated as follows:

$$\text{moisture content} = \frac{\text{Mwet} - \text{Mdry}}{\text{Mdry}} \times 100 \tag{1}$$

The procedure for obtaining the MC level of a specimen was per ASTM D4442 [21]. Method A consisted of the basic oven drying method according to ASTM D4442-15 to obtain MC values for the resulting samples. The initial weight of duplicate samples was used to calculate the average weight before oven drying. The samples were then placed in an oven heated to 103 ± 2 °C (214 to 221 °F) until no significant weight change was observed during the 4-hour weighing period, as shown in Figure 2. Table 5 shows the obtained MC levels of the as-received specimens. Samples were cut into different shapes and pieces due to the requirement of each test, as shown in Figures 3 and 4. This study used three nominal MC levels (5%, 10%, and 15%) to test the effect of MC levels on thermal degradation and flammability efficiency.

**Table 5.** Specimen MC level (as received).

| Specimen and MC Level Per ASTM D4442-15 (%) (Seven Replicates for Each Material) | | | | | | | | |
|---|---|---|---|---|---|---|---|---|
| Material/Replicate | 1 | 2 | 3 | 4 | 5 | 6 | 7 | AVE (%) |
| A | 14.1 | 14 | 13.7 | 13.9 | 8.7 | 13.6 | 13.8 | 13.5 |
| B | 18 | 11 | 7.8 | 11 | 11.4 | 11 | 10.7 | 11.7 |
| C | 4.7 | 4 | 4.3 | 4.4 | 4 | 4.3 | 4.7 | 4.5 |
| D | 3.1 | 2.7 | 3 | 3.2 | 3.3 | 3 | 3.2 | 3.2 |
| E | 6 | 6.3 | 5.8 | 6.2 | 5.9 | 6 | 5.8 | 6 |
| F | 7 | 7.4 | 6.9 | 7.2 | 7.5 | 6.8 | 7 | 7.2 |
| G | 5.5 | 5.6 | 6 | 5.9 | 5.5 | 5.6 | 5.4 | 5.7 |

TGA was performed at three different heating-rate levels (5, 15, and 25 °C/min) so that the thermal degradation kinetics of fuels would be investigated over a range of heating conditions. The heating rate is known to affect the position of the TGA curve and the maximum degradation rate. Thermal conductivity measurements were conducted for three replicates in 25 and 100 °C so that the fuel's conductivity and diffusivity properties could be quantified in a range of temperatures.

The flammability test using the cone calorimeter used three replicates at each heat-flux level for each fuel MC level. The cone calorimeter is a flammability test device based on the direct relationship between the heat released during flammability and the oxygen consumption. The heat generated is directly related to the intensity of the fire.

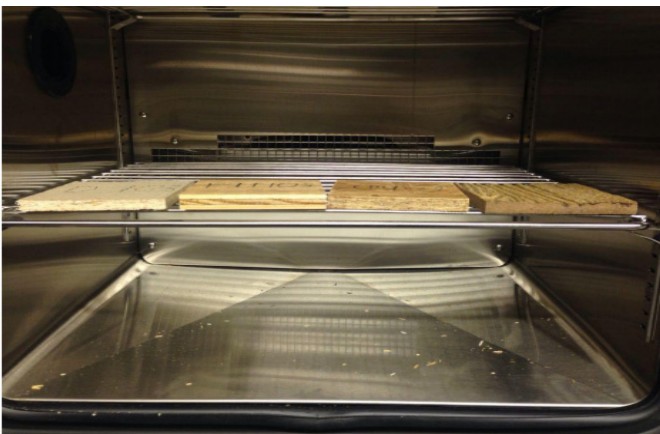

**Figure 2.** Samples in the environmental chamber.

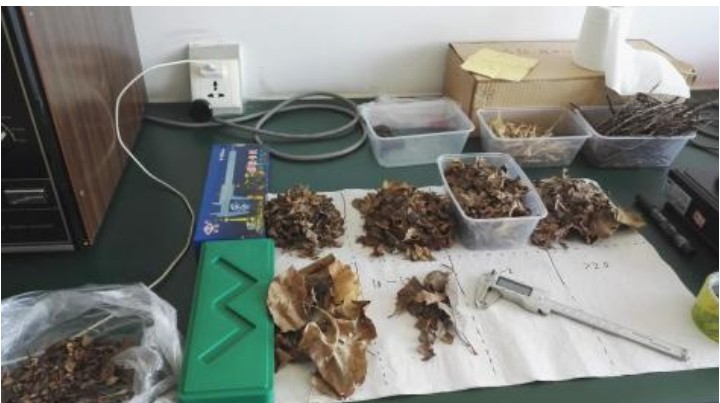

**Figure 3.** Sample material and measurement for pyrolysis test.

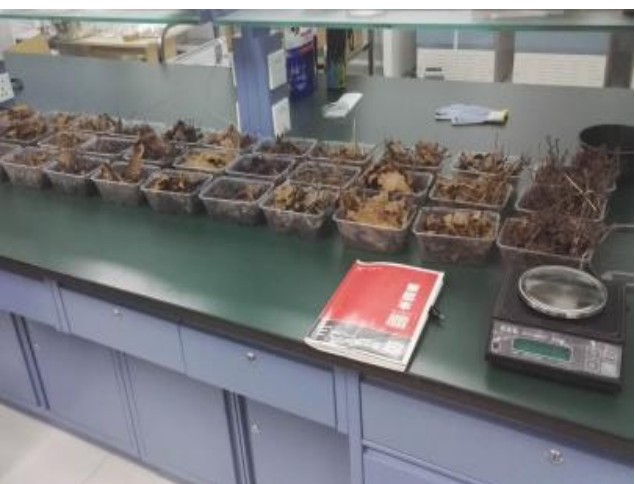

**Figure 4.** Sample material and weighing for pyrolysis test.

To maintain the material's flammability, it was exposed to external sources of thermal radiation. Evaporation of the moisture contained in the fuel dilutes the flammability gases, lowers the temperature of the flammability zone, and produces more unburned white smoke. Incompletely flammable products such as carbon monoxide (CO), nitrogen oxides (NO), and particulate matter (PM 2.5) increase the wet flammability of fuels [22]. Since the critical heat flux depends on the fuel type and MC level, each fuel was tested to determine the appropriate heat flux levels for this type of fuel according to its combustion results from the test at the critical heat-flux level. The developing stage of fire growth is characterized

by an external heat flux (around 20–60 kW/m²), so we selected 20 kW/m², 30 kW/m², and 50 kW/m² as our cone calorimeter heat-flux values [23]. Each sample was cut into 10 cm × 10 cm × 12.5 mm in the cone calorimeter, as shown in Figure 5.

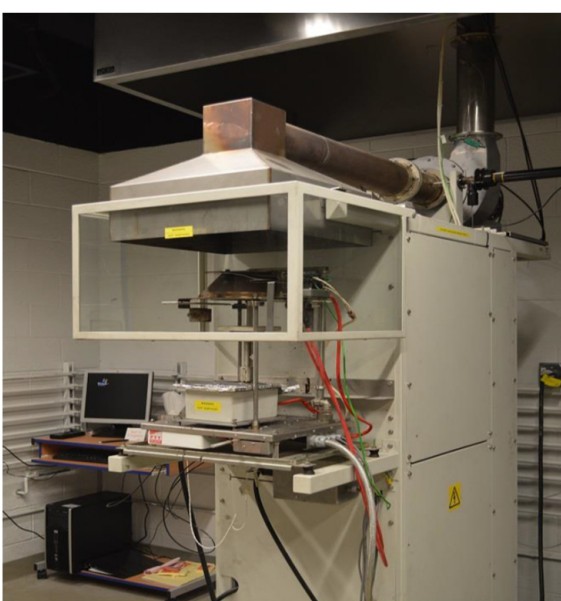

**Figure 5.** Cone calorimeter.

The activation energies measured by TGA of different pyrolysis phases can be calculated by pyrolysis kinetics [24]. To study the pyrolysis kinetics of woody biomass, weight loss generated from TGA could be used as a function of temperature, which has been proven in previous studies. TGA was conducted at three different heating-rate levels (5, 15, and 25 °C/min). In this way, the rate of thermal decomposition of fuels under specific heating conditions can be studied. TA instruments' flow meter measures the thermal conductivity of selected timbers. Experiments were performed at two temperature levels (25 °C and 100 °C) for each MC level according to the ASTM C 518 procedure [25]. Dynamic parameters were calculated by linear regression [26].

To measure thermal conductivity, pellet samples were cut from properly conditioned cylindrical samples and subjected to three heat-flux levels (5, 15, and 25 K/min) at 25 and 100 °C in a nitrogen atmosphere. A purge rate of 20 mL/min was tested to quantify the electrical conductivity and diffusion properties of the fuel over a range of temperatures. Thermal decomposition kinetics, preexponential coefficient (A), and activation energy (E) can be obtained from Equation (1), for which at each HR, $\alpha$ represents fraction reacted (or conversion factor):

$$\frac{d\alpha}{dT} = A(1-\alpha)\exp\left[-\frac{E}{RT}\right]/\beta \tag{2}$$

$\alpha$ = fraction reacted (dimensionless), $A$ = pre-exponential factor (min⁻¹), $\beta$ = heating rate (K/min), $E$ = activation energy (J/mol), $R$ = gas constant (=8.316 J/(mol K)), $T$ = absolute temperature (K), exp = Euler's number exponential, and $d\alpha/dT$ = rate of change of $\alpha$ with T.

In this study, the method of Ozawa, Flynn, and Wall (model-free iso-conversional methods) are explored, assuming that the conversion is constant, and the rate constant depends on the temperature. Kinetic parameters (*E* and *A*) were determined using different modeling and analysis techniques, such as differentiation, integration, and approximation [27]:

$$E = \left(\frac{R}{b}\right)\Delta\log(\beta)/\Delta\left(\frac{1}{T}\right) \tag{3}$$

$$A = \left(-\frac{\beta R}{E}\right)[\ln[1-\alpha]]10^{\alpha} \tag{4}$$

At different heat rates, linear regression analysis was used to calculate $\Delta \log(\beta)/\Delta(1-\text{T})$ by using a point of constant conversion from a series of decomposition curves.

A first approximation of activation energy ($E$) was obtained at the assumption that an initial value of b = 0.457; then, a new value of b could be determined by using the first approximation $E$. After $E/RT > 60$, MATLAB was used to analyze the data above, obtaining the tendency curve of a and b and estimating a and b. For first-order reactions ($n = 1$), the value of the curve can be determined as follows:

$$g_{(\alpha)} = \int_0^{\alpha} \frac{d\alpha}{1-\alpha} = -\ln(1-\alpha) \; n = 1 \tag{5}$$

Next, we continue the iterative process until the activation energy value does not change in the next iteration.

For the fuel properties test, 46 replicates were performed for the sample MC value and 63 replicates for the selected material size test; for the pyrolysis performance test, 189 replicates were used for the TGA test, and 42 replicates were used for the thermal conductivity test. In the flammability test, 189 replicates were tested using a cone calorimeter.

Quantitative data analysis was performed for the small-scale thermal analysis, fuel wildland property measurements, and flammability tests, focusing on understanding the following relationships:

(1)  Thermal property (*TP*) as a function of fuel type (*FT*), wildland property (*SP*), and MC, represented as $TP = f_1(FY, SP, MC)$;
(2)  Flammability property (*FP*) as a function of *FT*, *SP*, and *MC*, represented as $TP = f_2(FY, SP, MC)$;
(3)  The relationship between functions $f_1$ and $f_2$.

Linear and non-linear correlation analysis was performed to test our hypothesis that wildland timber characteristics can be estimated using thermal and flammability properties of the fuel.

It has been reported in American wildland firefighters that 66% of injuries during wildfire suppression (between 2003 and 2007) were heat burns [28]. Wildland firefighters work under adverse environments (e.g., heat and fire exposure), which contribute to increasing heat strain. The harm of heat flux to the human body is mainly expressed by the different degrees of harm to the human body caused by different thermal radiation fluxes. The radius of injury includes first-degree burn (slight), second-degree burn (severe), and death radius, as described by the heat radiation effect model proposed by Morten Gamst Pedersen, which was used herein [29]. In the process of flammability, the surface of the flame emits radiant energy to the outer spaces at high temperature, which is used to measure the effects of thermal radiation on human health. The probability of injury to a person exposed to thermal radiation is related to the exposure time and heat flux.

$$P = -36.38 + 2.56 \ln\left(tq^{\frac{4}{3}}\right) \tag{6}$$

*P*—probability of injury to persons;
*t*—exposure time to persons, s;
*q*—heat radiation flux received by the human body, w/m$^2$.

The probability of injury to persons and percentage of death [28] are expressed as follows:

$$D = \int_{-\infty}^{P-5} \frac{1}{\sqrt{2\pi}} \exp\left(-\frac{u^2}{2}\right) du \tag{7}$$

where
*D*—percent death;
*u*—integral variable.

The equation represents the relationship between the probability density and the cumulative distribution function. The probability p of an individual being injured is a measure of the percentage D of death. Their numerical correspondence can be a conveniently normal distribution. When P = 2.67, D = 1%; when p = 5.00, D = 50%; when P = 8.06, D = 99.9% [29]. The harm of different heat-flux values to the human body is noted in Table 6.

**Table 6.** The harm of different heat-flux values to the human body [29].

| Heat Flux (kW/m$^2$) | Types of Human Injury |
|---|---|
| 37.5 | 100% of people die in 1 min, 1% in 10 s |
| 25.0 | 100% of people die within 1 min and are severely burned within 10 s |
| 12.5 | 1% of people die in 1 min and one-degree burns in 10 s |
| 4.0 | Causes pain for more than 20 s but does not blister |
| 1.6 | Long-term contact with no discomfort |

In seven of the wildfires analyzed, the absolute heat-flux peaks achieved values above 20 kW/m$^2$, which meant a dangerous exposure [30]. The effective duration of heat exposure was calculated when positive heat flux was recorded. The weight of heat exposure was calculated as the ratio of exposure time to the total time of work in the wildfire-suppression area. The thermal dosage for each exposure class recorded in the sensors inside the protective clothing was calculated using the heat flux and the exposure time using Equation (8) to assess the potential burn injury [31]:

$$TDU = (q_{in})^{\frac{4}{3}} \times t \tag{8}$$

*TDU* is Thermal Dosage Units [(kW· m$^{-2}$)$^{4/3}$ ·s], $q_{in}$ is the incident heat flux (kW· m$^{-2}$) and *t* is the exposure duration (s).

In the cone calorimeter test, 20 kW/m$^2$, 30 kW/m$^2$, and 50 kW/m$^2$ were selected. Based on the reference above, 20 kW/m$^2$ could be regarded as the injured heat flux that firefighters suffered from the forest fire, which will be used to calculate the corresponding time for firefighters to put out the fire or change the forest-fire-extinguishing target.

## 3. Results and Discussions

### 3.1. Data Analysis

The subsequent data analysis was carried out for the above typical materials in order to obtain basic pyrolysis indices parameters and flammability properties of the selected wildland fuels at three pre-determined MC levels and three pre-determined heating conditions. Table 7 shows thermal conductivity, activation energy, and ln(A) in different MC levels and heating conditions. According to the principle of oxygen consumption, the heat release of oxygen consumption per unit mass has little to do with the type of fuel, and its value is 13.1 MJ/kg, with an error of 5%. Mass-loss rate, heat-release rate, and efficient heat of combustion could be quantified by the collection of cone calorimeter test, and the CO and $CO_2$ analyzer could measure soot yield, and CO and $CO_2$ and a laser beam in the exhaust pipe analyzes the development of smoke by measuring its attenuation.

### 3.1.1. Fuel Properties

With normal changes in relative humidity, the dimensional changes in dry wood are small. Wetter air will cause slight expansion, and dry air will cause slight contraction. The changes in mass and dimensions of the wood or wood-based composite will affect the bulk density of the material. The test uses MC-grade density (i.e., the density based on the weight of a sample with water and its volume at the same water content) (ASTM D2395-14). The fuel property test parameters include thickness and density.

The wildland fuel density is determined by dividing the mass of the sample (in grams) by the volume (cc) at a given MC value. Figures 6 and 7 show the normalized thickness of wildland timbers as a function of three specific MC values. Figures 8 and 9 show the bulk

density of the materials as a function of MC levels. Experimental errors are highlighted in the figures.

**Table 7.** TTI (time to ignition) of the selected fuels.

| TTI (s) | HF (kW/m$^2$) | 20 | 30 | 50 |
|---|---|---|---|---|
| A | 5% | 192 | 34 | 24 |
|   | 10% | 357 | 71 | 25 |
|   | 15% | 703 | 65 | 31 |
| B | 5% | 233 | 32 | 10 |
|   | 10% | 307 | 56 | 15 |
|   | 15% | 465 | 49 | 17 |
| C | 5% | 192 | 70 | 24 |
|   | 10% | 258 | 85 | 25 |
|   | 15% | 317 | 101 | 31 |
| D | 5% | 329 | 65 | 21 |
|   | 10% | 406 | 142 | 31 |
|   | 15% | 625 | 119 | 33 |
| E | 5% | 301 | 56 | 20 |
|   | 10% | 343 | 98 | 29 |
|   | 15% | 461 | 104 | 36 |
| F | 5% | 191 | 32 | 13 |
|   | 10% | 210 | 77 | 22 |
|   | 15% | 306 | 76 | 32 |
| G | 5% | 206 | 63 | 20 |
|   | 10% | 206 | 97 | 26 |
|   | 15% | 375 | 115 | 30 |

The average thickness of material A was reduced from MC 5% to 10% and increased from MC 10% to 15%. The average thickness of material B increased with increasing MC. The average thickness of material C increased with the increase of MC, the average thickness of material E increased with the increase of MC, and the average thickness of material F increased with the increase of MC. The average thickness of material G decreased from MC 5% to 10% but increased from MC 10% to 15%.

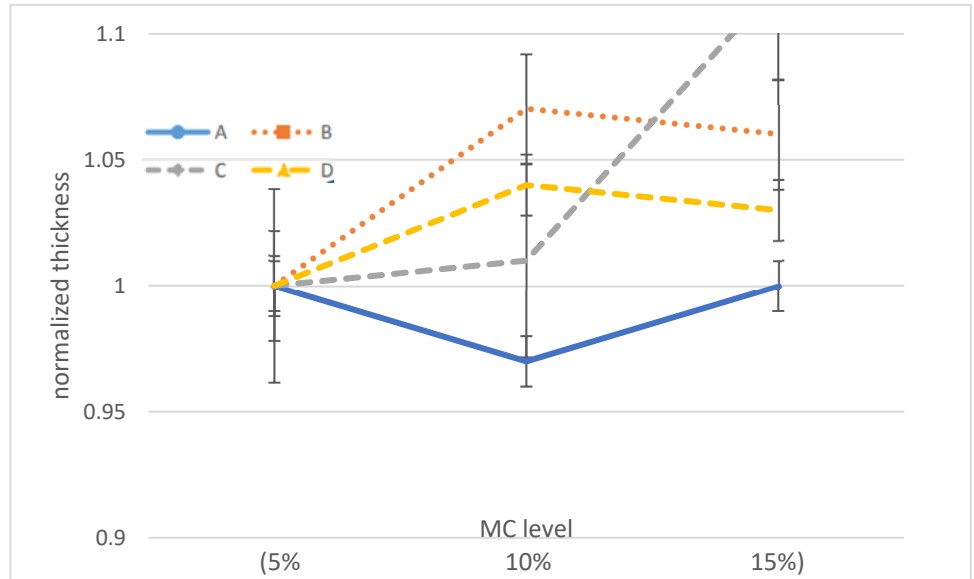

**Figure 6.** Normalized thickness (materials A, B, C, and D) vs. MC.

The densities of materials C, D, and E are close to each other. Material G has the highest density, and material B has the lowest density. As the MC level increased, the densities of materials A, D, E, and F increased slightly, while those of materials B and C decreased slightly. Material G has the highest density at 10% MC level.

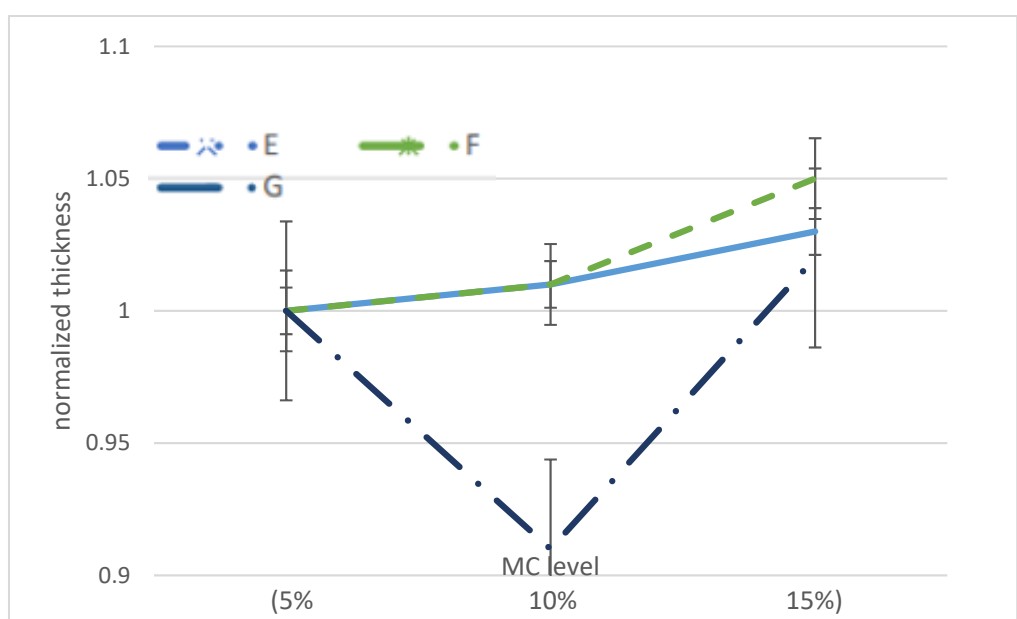

**Figure 7.** Normalized thickness (materials E, F, and G) vs. MC.

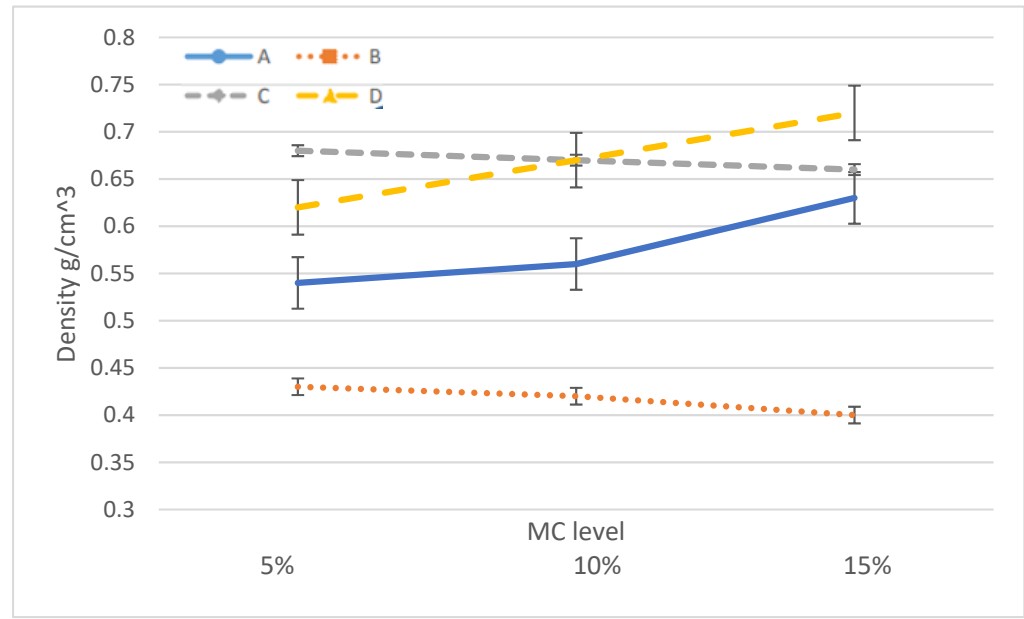

**Figure 8.** Density of selected wildland fuels (materials A, B, C, and D) vs. MC.

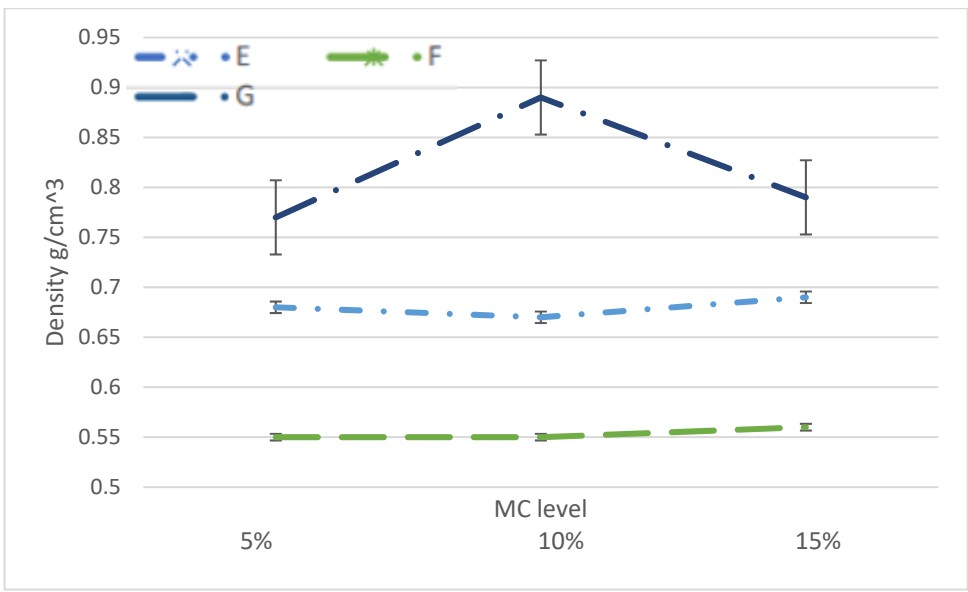

**Figure 9.** Density of selected wildland fuels (materials E, F, and G) vs. MC.

### 3.1.2. Pyrolysis Properties

For pyrolysis properties, there were three controlling factors (MC level, temperature, and heating rate) for the value of thermal conductivity, with E (activation energy) and A (pre-exponential factor) as parameters.

Thermal Conductivity

Figures 10 and 11 show the normalized thermal conductivity of the materials as a function of MC levels. Experimental errors are highlighted in the figures.

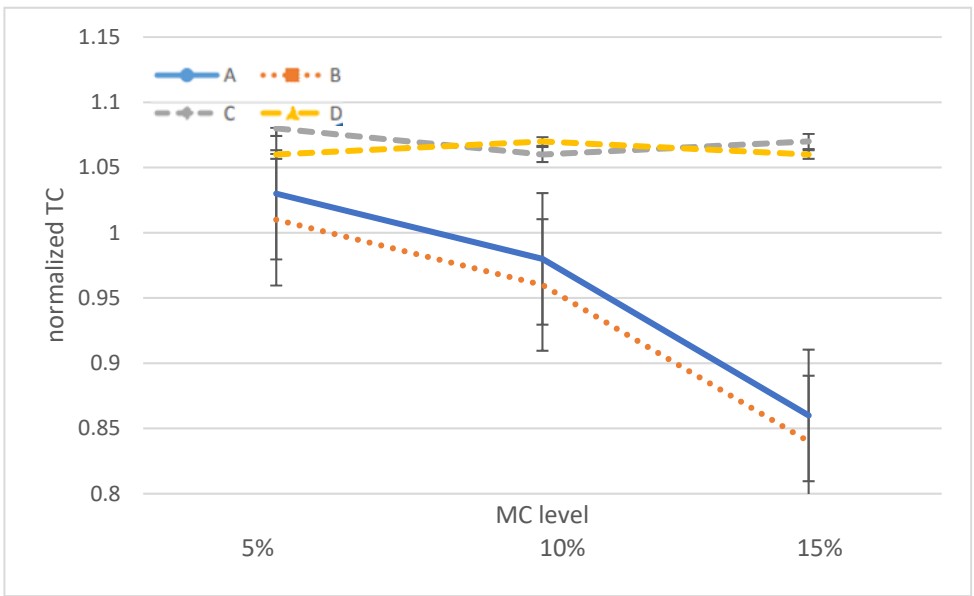

**Figure 10.** Normalized thermal conductivity of selected wildland fuels (materials A, B, C, and D) vs. MC.

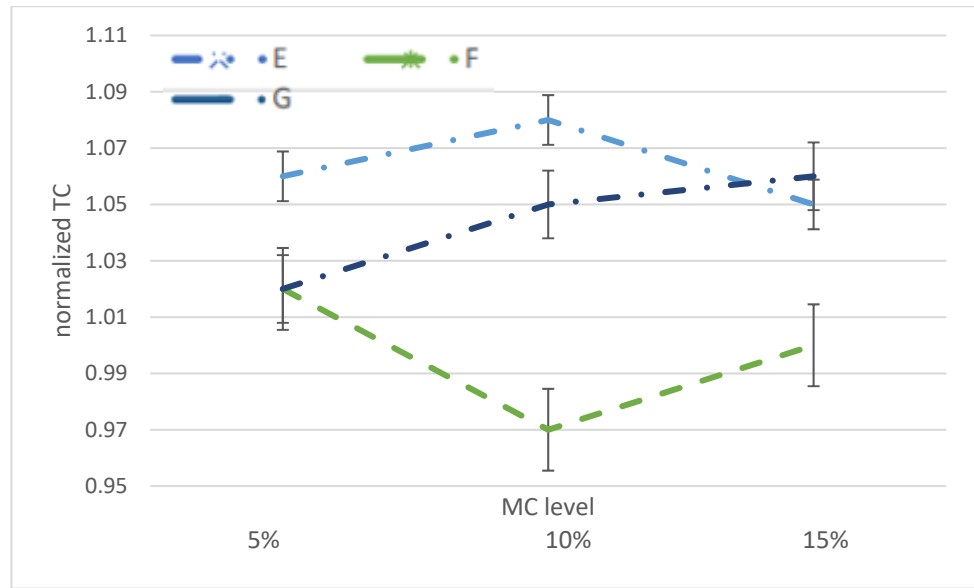

**Figure 11.** Normalized thermal conductivity of selected wildland fuels (materials E, F, and G) vs. MC.

The effect of temperature on thermal conductivity is less than that of MC level. As the temperature increases from 25 °C to 100 °C, the thermal conductivity increases in all materials except A, B, and F.

Pre-Exponential Factor (A)

Figures 12–18 show the pre-exponential factor (A) of the Arrhenius equation as a function of MC levels for materials A–G. $\alpha$ is the fraction reacted (or pyrolysized) or conversion factor of the material. Ln(A) changes greatly at the initial stage of thermal degradation, while it tends to be stable after the conversion coefficient $\alpha$ is higher than 0.25. Both heating rate and MC level had an evident effect on the pre-exponential factor.

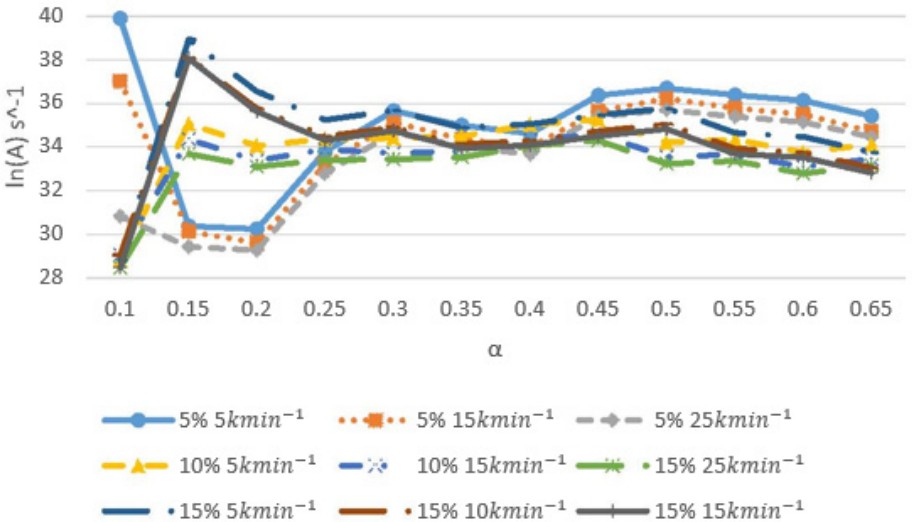

**Figure 12.** Material A ln(A) vs. MC.

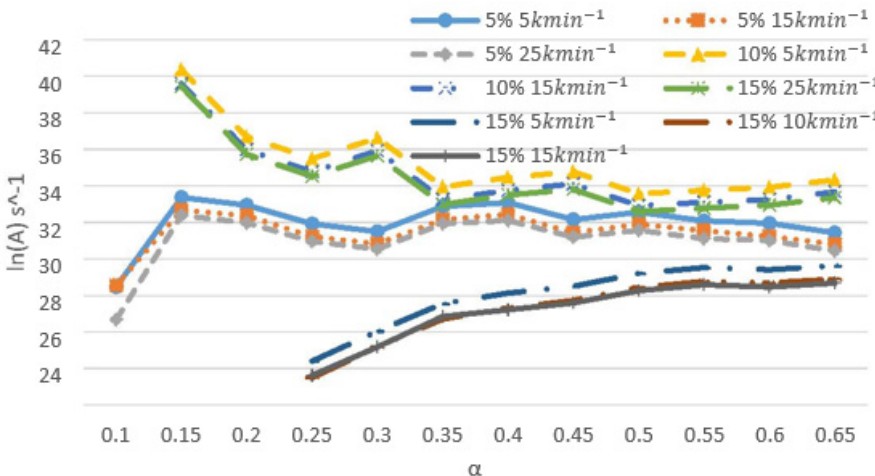

**Figure 13.** Material B ln(A) vs. MC.

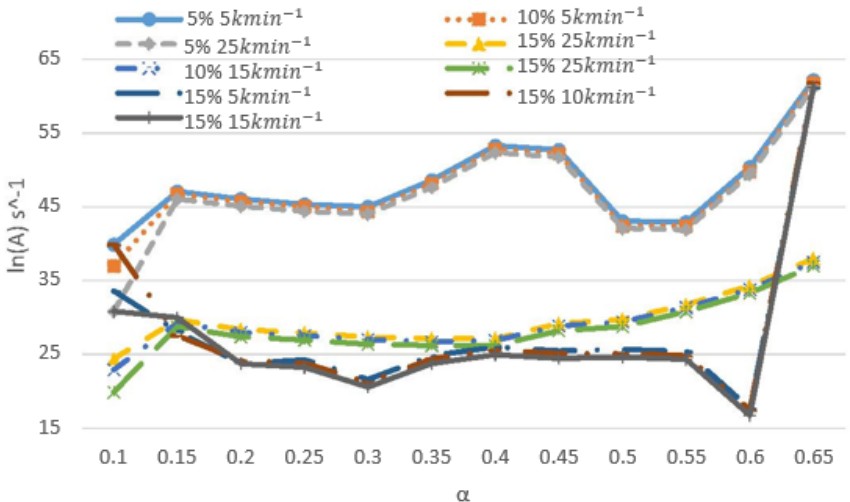

**Figure 14.** Material C ln(A) vs. MC.

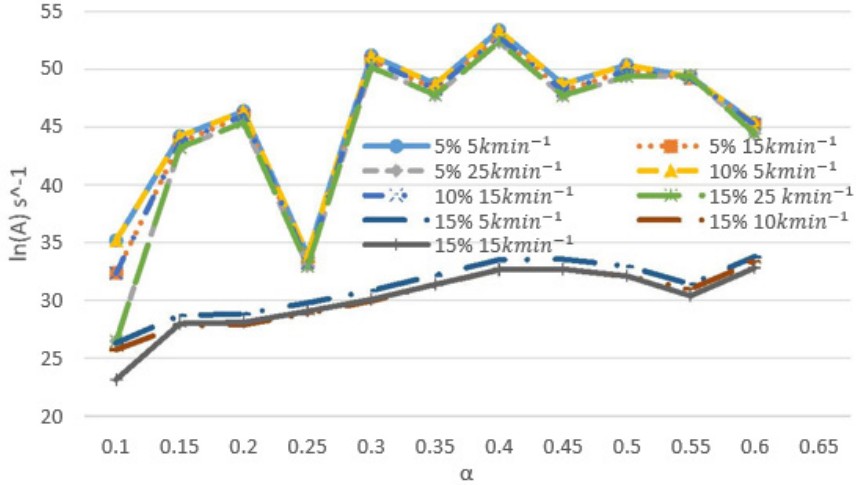

**Figure 15.** Material D ln(A) vs. MC.

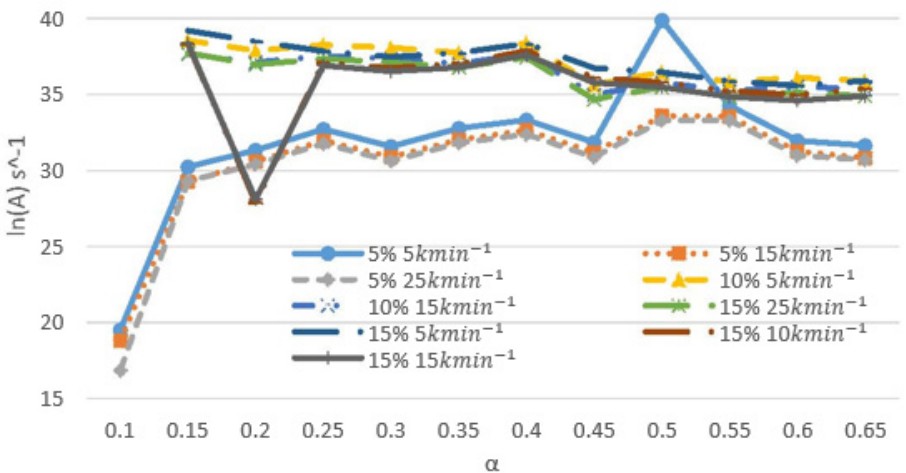

**Figure 16.** Material E ln(A) vs. MC.

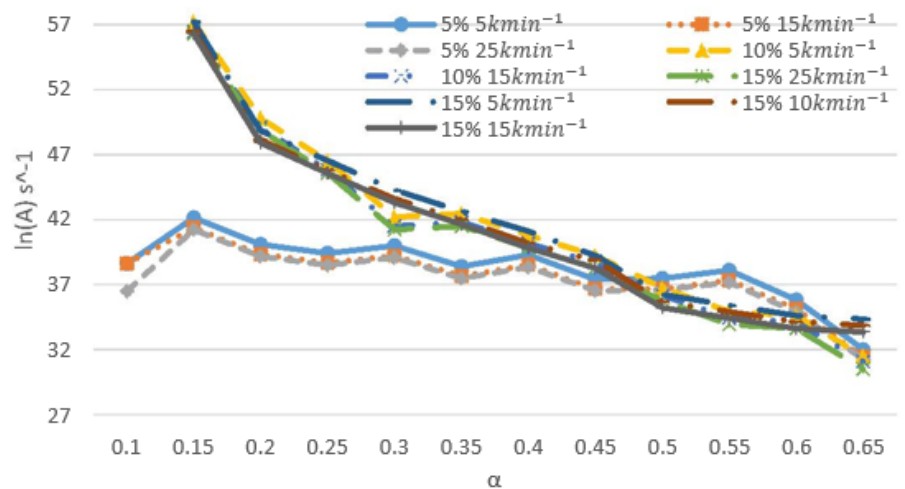

**Figure 17.** Material F ln(A) vs. MC.

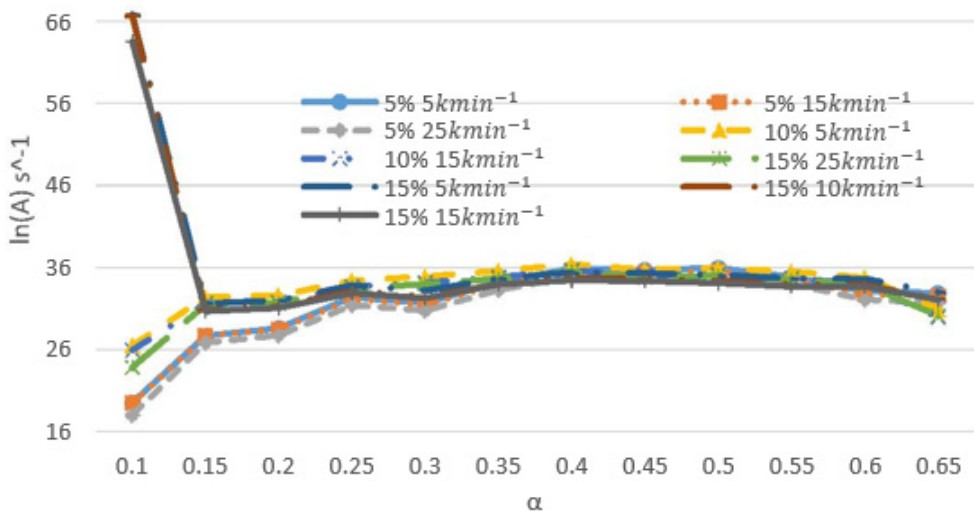

**Figure 18.** Material G ln(A) vs. MC.

Activation Energy (E)

Figures 19–25 show the activation energy (E) of wildland fuel as a function of the MC level. The activation energy value changes greatly in the initial pyrolysis period, then tends to be stable after the conversion coefficient α is above 0.25. This is comparable to the pre-exponential factor.

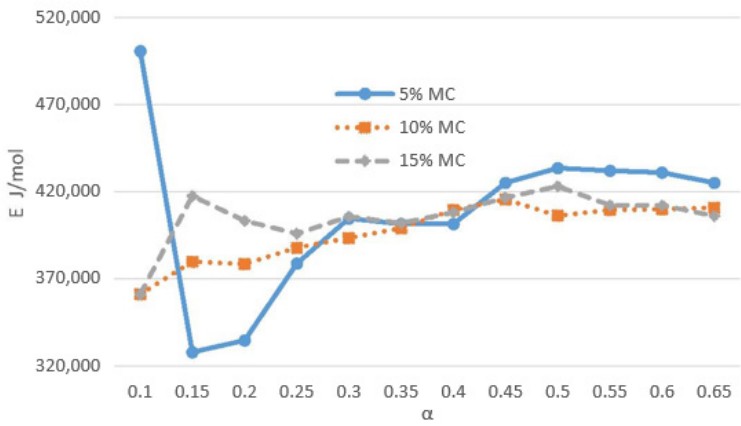

**Figure 19.** Material A activation energy vs. MC.

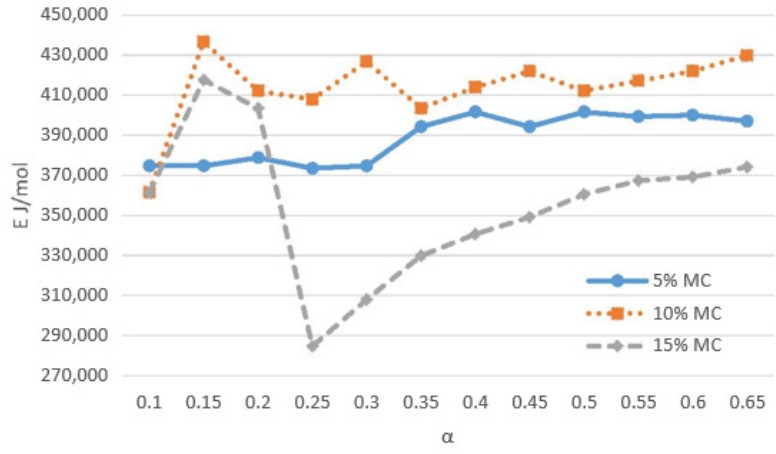

**Figure 20.** Material B activation energy vs. MC.

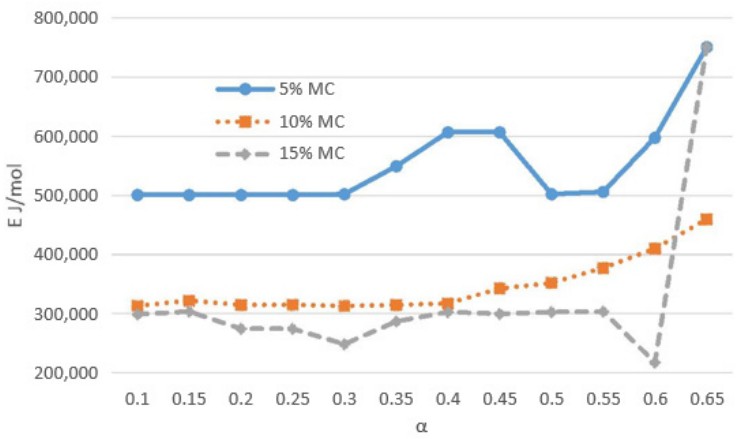

**Figure 21.** Material C activation energy vs. MC.

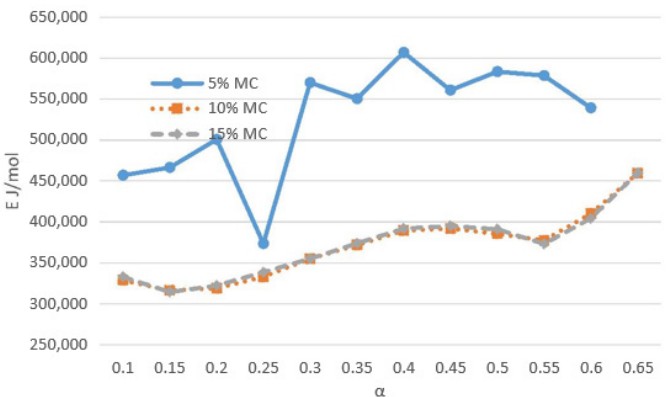

**Figure 22.** Material D activation energy vs. MC.

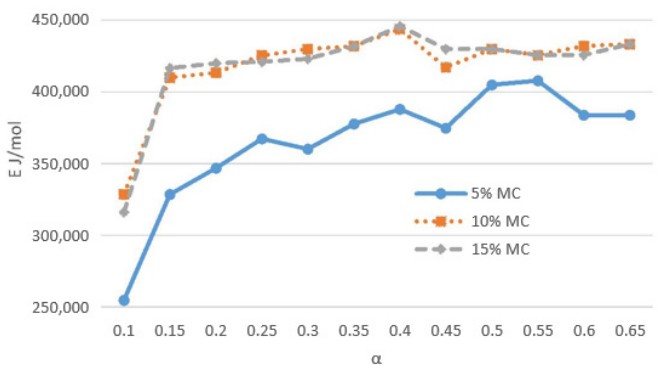

**Figure 23.** Material E activation energy vs. MC.

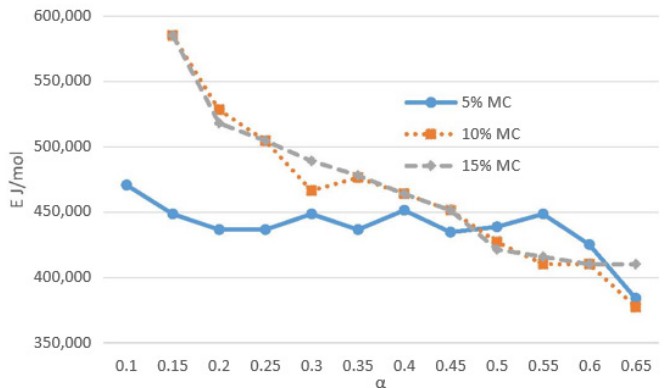

**Figure 24.** Material F activation energy vs. MC.

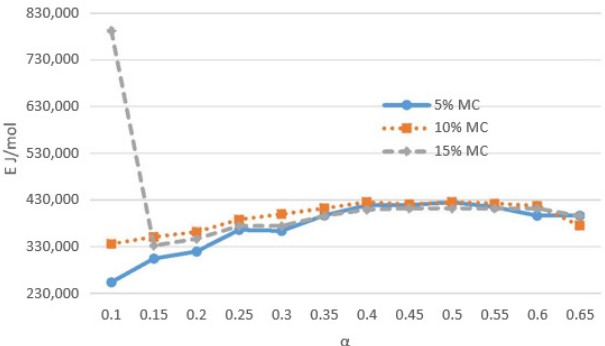

**Figure 25.** Material G activation energy vs. MC.

### 3.1.3. Flammability Properties

For flammability properties, there were two controlling factors (MC level and heat flux) for the values of ignition time, critical heat flux, peak heat-release rate (*pHRR*), effective heat of flammability (*EHC*), mass loss, and flameout time as parameters.

### Time to Ignition (*TTI*)

Time to ignition reflects the degree to which the material is ignited. The longer the *TTI*, the more fire-retardant the wood. Table 7 summarizes the *TTI* values (in seconds) of materials A–G at the three heat-flux levels (50, 30, and 20 kW/m$^2$) and three MC levels (15%, 10%, and 5%). Heat flux had a significant effect on *TTI*, and the higher the heat-flux levels, the smaller the *TTI* values. MC levels had significant effect at low heat-flux levels (30 or 20 kW/m$^2$).

### Critical Heat Flux (*CHF*)

The critical heat flux (*CHF*) values for ignition values of all materials were plotted and are shown in Figure 26. The CHFs of A and B increased as the MC level increased. The *CHFs* of C and D decreased as the MC level increased. The *CHFs* of E, F, and G increased from MC 5% to MC 10% but decreased from MC 10% to MC 15%.

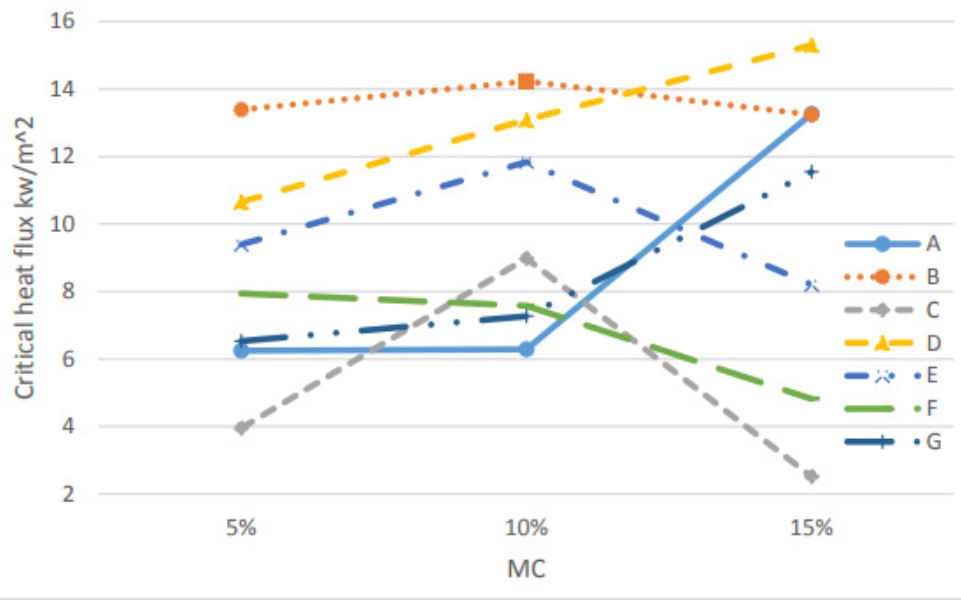

**Figure 26.** Critical heat flux of selected fuels vs. MC.

### Peak of Heat-Release Rate (*pHRR*)

*pHRR* is the most important input parameter for forest fire models, as it controls fire characteristics and indicates their contribution to fire evolution. The higher the maximum heat release rate of wood, the greater the heat release, the greater the risk of fire, and the greater the impact of fire on firefighters. The *pHRR* values for all materials are summarized in Table 8. The *pHRR* of all selected wildland materials increased as the heat-flux level increased and decreased as the MC level increased although with a few exceptions.

**Table 8.** *pHRR* of the selected fuels.

| *pHRR* (Kw/m²) | HF (kW/m²) | 20 | 30 | 50 |
|---|---|---|---|---|
| A | 5% | 160.21 | 182.8 | 176.68 |
| | 10% | 162.36 | 169.59 | 190.81 |
| | 15% | 131.83 | 158.48 | 200.75 |
| B | 5% | 174.88 | 192.64 | 230.36 |
| | 10% | 152.63 | 204.33 | 185.08 |
| | 15% | 112.59 | 168.43 | 182.84 |
| C | 5% | 193.95 | 206.76 | 303.76 |
| | 10% | 164.41 | 221.63 | 290.35 |
| | 15% | 166.38 | 179.91 | 251.81 |
| D | 5% | 255.25 | 242.35 | 339.85 |
| | 10% | 252.94 | 226.96 | 269.19 |
| | 15% | 191.85 | 168.31 | 221.19 |
| E | 5% | 301.01 | 192.61 | 289.66 |
| | 10% | 208.26 | 217.15 | 249.94 |
| | 15% | 213.52 | 201.33 | 253.8 |
| F | 5% | 162.09 | 188.32 | 319.2 |
| | 10% | 143.78 | 201.87 | 281.91 |
| | 15% | 155.35 | 151.95 | 203.15 |
| G | 5% | 170.17 | 217.05 | 370.89 |
| | 10% | 170.17 | 215.27 | 321.74 |
| | 15% | 141.66 | 187.42 | 387 |

Effective Heat of Combustion (*EHC*)

Effective heat of combustion (*EHC*) is the energy produced in a combustion reaction per unit mass of wood. The *EHC* values for all materials are summarized in Table 9. The MC level had a strong effect on the EHC of selected wildland fuels. HF levels had less effect on the *EHC* values.

**Table 9.** *EHC* of the selected fuels.

| *EHC* (MJ/kg) | HF (kW/m²) | 20 | 30 | 50 |
|---|---|---|---|---|
| A | 5% | 14.09 | 13.89 | 7.88 |
| | 10% | 14.32 | 14.12 | 12.23 |
| | 15% | 14.83 | 13.62 | 13.97 |
| B | 5% | 14.29 | 14.26 | 14.62 |
| | 10% | 13.70 | 13.10 | 13.89 |
| | 15% | 10.99 | 12.80 | 13.39 |
| C | 5% | 13.93 | 13.59 | 15.87 |
| | 10% | 13.89 | 13.87 | 13.50 |
| | 15% | 9.88 | 12.85 | 10.42 |
| D | 5% | 14.77 | 10.78 | 14.46 |
| | 10% | 14.46 | 10.40 | 14.22 |
| | 15% | 14.31 | 12.70 | 10.34 |
| E | 5% | 17.41 | 9.55 | 11.50 |
| | 10% | 18.32 | 17.37 | 18.71 |
| | 15% | 26.83 | 18.50 | 18.94 |
| F | 5% | 16.16 | 13.30 | 14.57 |
| | 10% | 13.58 | 13.88 | 14.06 |
| | 15% | 13.40 | 7.54 | 10.05 |
| G | 5% | 14.73 | 13.94 | 15.53 |
| | 10% | 14.73 | 13.24 | 12.30 |
| | 15% | 13.28 | 13.24 | 14.33 |

Mass Loss (*ML*)

*ML* can be used to estimate thermal degradation rates and resulting volatile concentrations. They have a great effect on flammability. *MLR* is measured with a load cell placed below the sample. The *ML* values for all materials are summarized in Table 10. The *ML* of the material increases with increasing HF levels. MC levels had little effect on *ML*.

**Table 10.** *ML* (mass loss) of the selected fuels.

| *ML* (Mass Loss) | HF (kW/m$^2$) | 20 | 30 | 50 |
|---|---|---|---|---|
| | 5% | 14.09 | 13.89 | 7.88 |
| A | 10% | 14.32 | 14.12 | 12.23 |
| | 15% | 14.83 | 13.62 | 13.97 |
| | 5% | 14.29 | 14.26 | 14.62 |
| B | 10% | 13.70 | 13.10 | 13.89 |
| | 15% | 10.99 | 12.80 | 13.39 |
| | 5% | 13.93 | 13.59 | 15.87 |
| C | 10% | 13.89 | 13.87 | 13.50 |
| | 15% | 9.88 | 12.85 | 10.42 |
| | 5% | 14.77 | 10.78 | 14.46 |
| D | 10% | 14.46 | 10.40 | 14.22 |
| | 15% | 14.31 | 12.70 | 10.34 |
| | 5% | 17.41 | 9.55 | 11.50 |
| E | 10% | 18.32 | 17.37 | 18.71 |
| | 15% | 26.83 | 18.50 | 18.94 |
| | 5% | 16.16 | 13.30 | 14.57 |
| F | 10% | 13.58 | 13.88 | 14.06 |
| | 15% | 13.40 | 7.54 | 10.05 |
| | 5% | 14.73 | 13.94 | 15.53 |
| G | 10% | 14.73 | 13.24 | 12.30 |
| | 15% | 13.28 | 13.24 | 14.33 |

Flameout Time (*TTF*)

Flameout time reflects the degree to the fire decay. The longer the *TTF* is, the longer the duration that the wood fire lasts. The flameout time values for all materials are summarized in Table 11. The *TTFs* of all materials increased with increasing MC levels and, with few exceptions, decreased with increasing heat flux.

By clarifying the property variables results for each material, although different material acts differently in fuel, pyrolysis, or flammability, some similarities (bold text) can still easily be summarized from the Table 12.

Comparing with research results of other scientists and analysis of physical and chemical special features, similar conclusions (bold text) can be found in Table 13, which proves the validation of the research results.

This study measured the pyrolysis and flammability of some selected wildwood fuels and found that some control factors (MC value, heating conditions) influenced the outcome variables, especially wildwood flammability. This research advances our understanding of fuel properties in small-scale wildland fire. Pyrolysis analysis improves the understanding of reaction mechanisms under different heating conditions and the importance of fuel blending in fuel conversion and reaction rates.

**Table 11.** Flameout time of selected fuels.

| *TTF*(s) | HF (kW/m$^2$) | 20 | 30 | 50 |
|---|---|---|---|---|
| A | 5% | 1443.3 | 1153 | 868.3 |
| | 10% | 1616.7 | 1090 | 981.7 |
| | 15% | 1950 | 1406.3 | 1004.7 |
| B | 5% | 1486 | 1228 | 844 |
| | 10% | 1479 | 1304.7 | 1015.7 |
| | 15% | 1856 | 1346.7 | 1075.7 |
| C | 5% | 1168 | 706 | 402 |
| | 10% | 1159 | 751 | 746 |
| | 15% | 1423 | 992 | 802 |
| D | 5% | 998 | 818.3 | 441.3 |
| | 10% | 1109.7 | 876.3 | 597 |
| | 15% | 1363.7 | 953.7 | 691 |
| E | 5% | 1917.7 | 1558.7 | 1197 |
| | 10% | 1943.3 | 1713.3 | 1408 |
| | 15% | 1716 | 1764.3 | 1740.3 |
| F | 5% | 1027 | 805 | 469 |
| | 10% | 1217.3 | 822 | 566.3 |
| | 15% | 1112 | 441.7 | 1714.7 |
| G | 5% | 1312 | 924.3 | 864.3 |
| | 10% | 1347.2 | 932 | 715.7 |
| | 15% | 1620 | 1056 | 716.3 |

**Table 12.** Properties of selected wildland fuels (fuel, pyrolysis, and flammability properties).

| Fuel Properties | Pyrolysis Properties | Flammability Properties |
|---|---|---|
| The densities of materials C, D, and E are close to each other. Material G has the highest density, and material B has the lowest density. As the MC level increased, the densities of materials A, D, E, and F increased slightly, while those of materials B and C decreased slightly. Material G has the highest density at 10% MC level. | **The effect of temperature on thermal conductivity is less than that of MC level.** As the temperature increases from 25 °C to 100 °C, the thermal conductivity increases in all except materials A, B, and F. **The *ln(A)* values varied in the early stage of pyrolysis but appeared to be more stable when the conversion factor $\alpha$ was 0.25 or higher. Both MC level and heating rate had strong effect on the pre-exponential factor.** *E* values varied in the early stage of pyrolysis but appeared to be more stable when the conversion factor $\alpha$ was 0.25 or higher. This is similar to pre-exponential factor. | **Heat flux had significant effect on *TTI*; the higher the heat flux levels, the smaller the *TTI* values.** MC levels had a significant effect at low heat-flux levels (20 or 30 kW/m$^2$). The *CHFs* of A and B increased as the MC level increased. The *CHFs* of C and D decreased as the MC level increased. The *CHFs* of E, F, and G increased from MC 5% to MC 10% but decreased from MC 10% to MC 15%. **The *pHRR* of all selected structural materials increased as the heat flux level increased and decreased as the MC level increased although with a few exceptions. The MC level had a strong effect on the *EHC* of selected wildland fuels. HF levels had less effect on the *EHC* values. The *ML* of the material increases with increasing HF levels. MC levels had little effect on *ML*. The *TTFs* of all materials increased with increasing MC levels and, with few exceptions, decreased with increasing heat flux.** |

**Table 13.** Comparison with research results of other scientists and analysis of physical and chemical special features.

| Author/<br>Research Institute | Fuel and Findings |
|---|---|
| Pacific Southwest Forest and Range Experiment Station (1976) | The greater thermal conductivity and heat capacity of dense fuels acts to increase the amount of heat needed for ignition and the fuel ignition time. As the moisture content increases, the ignition time increases. |
| Building and Fire Research Laboratory (BFRL), Gaithersburg, USA (2020) | OSB and wood studs (re-entrant corner wall assemblies). Under the same windspeed, the slope of the relationship of the mass of flammable timbers fuel and the projected area turned out to be the same, which was affected by the wind speed within the setting's experimental range. |
| Oregon State University, USA (2020) | Douglas fir (Pseudotsuga menziesii), western juniper (Juniperus occidentalis), ponderosa pine (Pinus ponderosa), and grand fir (Abies grandis). Under different combinations of the species above and moisture content, the size of flammable wildland woods to ignite spot fires was explored. |
| State Key Laboratory of Fire Science, Hefei, China (2020) | 600–1100 °C: Pine needles, 6–14 mm steel spherical particle. (1) As particle temperature and radiant heat flux increase, the likelihood of ignition increases. (2) As particle size and temperature increase, the critical radiative heat flux required for ignition decreases. The ignition-delay time decreases with increasing radiative heat flux. |
| Auburn University, AL, USA (2020) | Pinus ponderosa and Pinus monticola needle fuel beds. Fuel charge has a strong positive effect on flame height, specific surface area, and flame depth. After adding fuel, the flame intensity and temperature do not change significantly. |
| Huali Hao (2020) Massachusetts Institute of Technology, USA | Dried oak, larch, and red cedar. The effect of heat flux on the fire performance of oak, larch. and red cedar is attenuated with its increment. An increased heat flux promotes heat release, charring rate. and $CO_2$ production, while it has a negative effect on CO release. The peak CO production of oak is initially decreased and then increased with the rise of heat flux, whereas the peak CO production of larch is first increased and then decreased. |

To study the whole process of firebrands requires an understanding of the principles of flammability, aerodynamics, fire science, and meteorology and their application in environments that may, at best, be poorly understood or quantified. In addition, knowledge is required of the botanical features of wood bark, including shape, density, and shedding habit, since these can be expected to affect flammability and aerodynamic behavior. A full review of all the above fields was not possible here, and this study addresses only those principles that apply to a laboratory study. The selected wildland fuel collection could not include all potential different types due to different MC level, seasons, and sunlight. The flammability test could not consider the wind speed effect on the ignition due to the lack of wind tunnel equipment. The replicates of each test may not be sufficient due to the huge numbers of tests considering different effect factors and much more work, including collecting varieties and cutting timbers. Weather data are lacking, which combined with topography and fuel elements to affect the formation of fire direction indicators are crucial to properly interpreting a wildfire's burn pattern.

The selected seven typical wildland fire risk classification is obviously distinguished from the Table 14 based on flammability test results.

1. For TTI, material F has a shorter ignition time, and material D has a longer ignition time;
2. A high value of thermal conductivity means a long ignition time, and material B and F have a shorter ignition time, and material D has a longer ignition time;
3. Moreover, materials A, B, and G have the lowest peak HRR, whereas material D has the largest peak HRR;

4. For MLR, materials B, E, and F have a higher MLR, and materials A and C have a lower MLR. A higher MLR indicates a higher pyrolysis rate with more flammable volatiles released, causing more heat release from their oxidation. Because the average charring rate is correlated to the MLR, materials B, E, and F have a higher charring rate.

**Table 14.** Wildland Fire Risk Classification (HF: 20 kW/m$^2$, MC 5%).

| Material | TTI | pHRR | Flameout Time | ML (Mass Loss) | Fire Risk Classification |
|---|---|---|---|---|---|
| A | 192 | 160.21 | 1443.3 | 14.09 | High |
| B | 233 | 174.88 | 1486 | 14.29 | High |
| C | 192 | 193.95 | 1168 | 13.93 | Intermediate |
| D | 329 | 255.25 | 998 | 14.77 | Low |
| E | 301 | 208.26 | 1917.7 | 17.41 | Intermediate |
| F | 191 | 208.26 | 1027 | 16.16 | High |
| G | 206 | 170.17 | 1312 | 14.73 | Intermediate |

The top fire-risk wildland materials include materials A, B and F; intermediate fire-risk wildland materials include materials C, E, and G; and low fire-risk wildland material includes material D.

## 4. Conclusions

Based on the above results and discussions, the following conclusions can be drawn: Material F (*Pinus massoniana* from Guangxi Province) and material B (*Eucalyptus robusta Smith* from Guangdong Province) are more easily ignited than other materials under the heating condition and MC level. Materials A, B, and F represent the greatest fire hazard, while material D represents less fire hazard compared with other species. Material F (*Pinus massoniana* from Guangxi Province) and material B (*Eucalyptus robusta Smith* from Guangdong Province) represent the greatest fire hazard, and material D (*Betula platyphylla Suk.*—birch from Dongbei Province) represents less fire hazard compared with other species. Both Guangxi and Guangdong Provinces are regarded as provinces with a warm climate and strong sunlight all year long regarding the selected seven materials, while Dongbei Province has with coldest climate and a lack of sunlight.

From the perspective of firefighting, the following aspects can be concluded to prevent and control forest wildland fire:

1. Forest fire barriers or fuel breaks should be used to separate *Eucalyptus robusta* Smith and *Pinus massoniana* before or in the fire due to their high risk for ignition and strong flammability; we recommend removing, controlling, and replacing high-risk flammable timbers with low-risk flammable timbers as a part of long-term wildland fire management strategies;

2. The fire commander should use some research to test conclusions and make an accurate analysis and judgment: The TTI time indicates the ideal time for firefighters to put out fire (329 s as maximum time for firefighters to put out early-stage forest fire for the selected wildland fuels example, while *Pinus massoniana* only allows 191 s), the peak of heat-release time indicates a fully developed fire to suggest firefighters finish work before the forest fire spirals out of control, and the flameout time (1917 s as maximum time for the selected wildland firebrand flammability before the decay of fire, while *Pinus massoniana* allows 1168 s) represents the moment of low risk of fuel ignition, so firefighters could allow the fuel to burn out and change the extinguishing target to other regions of developing forest firebrands.

**Author Contributions:** Conceptualization, W.Y., B.H.A.B. and L.G.; methodology, W.Y. and N.N.; formal analysis, H.M.; investigation, W.Y. and L.G.; resources, W.Y.; writing—original draft preparation, W.Y. and B.H.A.B.; writing—review and editing, W.Y. and B.H.A.B.; supervision, B.H.A.B.; project administration, W.Y. and B.H.A.B. All authors have read and agreed to the published version of the manuscript.

**Funding:** This research received no external funding.

**Institutional Review Board Statement:** Not applicable.

**Informed Consent Statement:** Not applicable.

**Data Availability Statement:** Data available on request from the authors.

**Conflicts of Interest:** The authors declare that they have no known competing financial interests or personal relationships that could have appeared to influence the work reported in this paper.

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
