# Peer review of "A Laboratory-Scale Study of Selected Chinese Typical Flammable Wildland Timbers Ignition Formation Mechanism"

_fire, doi:10.3390/fire6010020_

Round 1
Reviewer 1 Report
This paper focused on the flammable wild land timbers ignition formation mechanism in China. Unfortunately, it is poorly organized with many problems, and I cannot recommend publication in the current state .
Major comments are:
1. This paper is more like an experimental report to me. There is a major problem lacking innovation, so in the current state I cannot recommend publication.
2. Considering the repeatability of the tests, experimental errors should be stated, and they must be highlighted in the figures using error bar (especially in Figs. 10-12).
3. What types of the instruments and sensors used in Section 5? And what’s the precision?
4. Conclusions must be re-organized, which is not a good summary in the present form.
5. Table 1 is not displayed correctly.
6. Many typos are found. Please check carefully the formatting errors, including the equation, reference and citation, etc.
7. If Figs. 1-4 are cited from other papers, they must be cited correctly, and permissions are necessary from the authors.
Author Response
Dear reviewer:
Thanks very much for your comments and professional advice. These opinions help to improve academic rigor of our article. Based on your suggestion and request, we have made corrected modification on the revised manuscript. We hope that our work can be improved again.
Yours sincerely,
Wenxu Yang

Reviewer 2 Report
Article can be accepted after revision according following comments:
1. Highlights section can be improved using the main research results. Strict character requirements must be followed.
2. Graphical abstract can be added to manuscript.
3. It is advisable to expand and structure the review of modern achievements in the chosen field of research.
4. Research methods are not new. Many similar articles have been published over the past 5 years. Authors need to explain their uniqueness, or provide links to works from which the methodology for conducting studies is taken.
5. It is advisable to expand the introductory part of the article by dividing it into two independent parts: experiments; simulation.
6. It is advisable for the authors to expand the description of the main errors in the performed studies.
7. It is need to improved using comparison with research results of other scientists and analysis of physical and chemical special features.
Author Response

(The authors gave the same response as above.)

Reviewer 3 Report
The manuscript discusses the basic pyrolysis and flammability of wildland species in typical forest areas of China. The Authors described results of very interesting laboratory experiments and evaluated a number of significant parameters.
In my opinion, the topic is of interest from both scientific and practical points of view.
The manuscript could be recommended for publication in Fire.
However, the manuscript lacks a standard section structure. This makes it difficult to understand methods, results, and discussion. In my opinion, the text needs to be improved.
I believe that the article requires significant improvement in the form of presentation of the material. In some parts, it was difficult to read and understand the text, results, tables, and figures. I suggest that the authors review the following elements of the manuscript, namely:
1. Lines 36–114 Introduction: The introduction does not contain information about similar experiments and results in this area of research. Perhaps the information in Table 1 could be included in the Introduction section to show the current state of the issue.
2. Table 1 – Unfortunately, part of the table is hidden. Please fix it. Why “since 1970s”? There are review of papers of 2014–2020 only.
3. Lines 139: Is there a link to official data?
4. Lines 168–192: Why is the statement of the problem and the purpose of the study again given here?
5. Table 6: No explanation for table row with data "1-7". Please clarify.
6. Table 6, 7. I am not sure that the first column called “Ignition time (s)” is correct. As well, I am not sure that the first column called “PHRR (Kw/m2)” is correct! The same in the Table 9, 10, 11.
7. Figure 13–19 are difficult to read.
8. Lines 325–346 Conclusion: A large number of particular conclusions are given, which can be combined into a more general one. In addition, the number of abbreviations in the conclusions did not allow me to fully understand these results.
Author Response

(The authors gave the same response as above.)

Round 2
Reviewer 1 Report
Accept as revised.
Reviewer 2 Report
Article can be accepted.
Reviewer 3 Report
After reviewing the revised version of the text, I can confirm that the manuscript has been improved. The authors considered Reviewers comments and clarified some issues in the manuscript. In accordance with the corrections and additions made, the title of the manuscript was also changed to “Forest Firefighting Strategy Research Based on a Laboratory-scale study of Selected Chinese Typical Flammable Wildland Timbers Ignition Formation Mechanism”.
In my opinion, current version of the manuscript corresponds to the subject matter of the Fire journal and could be recommended for publication.